# Beating of a Spherical Liquid Crystal Elastomer Balloon under Periodic Illumination

**DOI:** 10.3390/mi13050769

**Published:** 2022-05-13

**Authors:** Wenyan Cheng, Quanbao Cheng, Changshen Du, Yuntong Dai, Kai Li

**Affiliations:** School of Civil Engineering, Anhui Jianzhu University, Hefei 230601, China; w_cheng@yeah.net (W.C.); cheng_quanbao@outlook.com (Q.C.); changshendu@yeah.net (C.D.); daiytmechanics@ahjzu.edu.cn (Y.D.)

**Keywords:** liquid crystal elastomer, beating, spherical balloon, periodic light, dynamic LCE model

## Abstract

Periodic excitation is a relatively simple and common active control mode. Owing to the advantages of direct access to environmental energy and controllability under periodic illumination, it enjoys broad prospects for application in soft robotics and opto-mechanical energy conversion systems. More new oscillating systems need to be excavated to meet the various application requirements. A spherical liquid crystal elastomer (LCE) balloon model driven by periodic illumination is proposed and its periodic beating is studied theoretically. Based on the existing dynamic LCE model and the ideal gas model, the governing equation of motion for the LCE balloon is established. The numerical calculations show that periodic illumination can cause periodic beating of the LCE balloon, and the beating period of the LCE balloon depends on the illumination period. For the maximum steady-state amplitude of the beating, there exists an optimum illumination period and illumination time rate. The optimal illumination period is proved to be equivalent to the natural period of balloon oscillation. The effect of system parameters on beating amplitude are also studied. The amplitude is mainly affected by light intensity, contraction coefficient, amount of gaseous substance, volume of LCE balloon, mass density, external pressure, and damping coefficient, but not the initial velocity. It is expected that the beating LCE balloon will be suitable for the design of light-powered machines including engines, prosthetic blood pumps, aircraft, and swimmers.

## 1. Introduction

Liquid Crystal Elastomer (LCE) is a responsive material synthesized from anisotropic rod-like liquid crystal molecules and long-chain stretch polymers [1,2,3,4,5,6], which can respond to external stimuli, such as thermal [7,8,9], electric [10,11], optical [12,13,14], magnetic [15] and chemical [16,17,18,19] fields. LCE generally enjoys the significant advantages of quick response to deformation, deformation recoverability and noiselessness [20], and it has broad application prospects in many fields, including artificial muscles [21,22,23], microsystems and MEMS [24,25,26,27], actuators and sensors [28,29,30,31,32,33], energy harvesters [34,35], medical devices [36,37], and telescopic optical devices [38,39].

The oscillation of plates and shells caused by electric, magnetic, optical, and thermal excitations is widely used in aerospace, military and civil engineering [40,41,42,43,44,45,46,47,48]. In recent years, light-powered oscillation of LCE has gained huge interests. It can convert light energy into mechanical energy for use in soft robots, micro factories and nanomachines without conventional motor drives [49,50,51,52]. The light-powered oscillation is caused by the large deformation of LCE due to the orientation sequence change in liquid crystal molecule [53,54,55,56,57]. For example, ultraviolet light can induce the photoisomerization of azobenzene molecules, which leads to the large deformation of LCE [58,59]. Finkelmann et al. synthesized LCE containing azobenzene molecules with a photoinduced contraction strain close to 20% [60]. Zhao et al. theoretically studied the forced oscillation of a photosensitive liquid crystal elastomer cantilever beam, and realized the control of the oscillation characteristics by adjusting the illumination period and rate [61]. Cheng et al. proposed a light-powered self-oscillating LCE balloon with self-shadowing coating, and investigated the dynamic process of the self-sustained oscillating LCE balloon based on the dynamic LCE model and ideal gas model [62]. 

Although many experiments and theoretical studies have been carried out to investigate the static deformation and self-sustained oscillation of LCE induced by static light sources, there are scant research works on the dynamic response of LCE under periodic illumination. In this paper, we propose a model for a spherical LCE balloon under periodic illumination, establish the governing equation of motion, and then further study the beating of the LCE balloon under periodic illumination. We can achieve the dynamic deformation and its deformation response time of the LCE balloon by controlling the light period and light time of periodic illumination. Theoretical calculations show that there exists an optimal lighting period and lighting time rate that maximizes the amplitude of the beating. We also extensively explore the effects of light intensity, contraction coefficient, amount of gaseous substance, volume of LCE balloon, mass density, external pressure, damping coefficient, and initial velocity on the amplitude of the beating. In Section 2, the theoretical model and the governing equation of motion for LCE balloon are introduced. In Section 3, the optimal illumination period and illumination time rate are analyzed, and the effects of each parameter on the beating amplitude of the system are discussed. In Section 4, the results are collated to draw the conclusions. 

## 2. Theoretical Model and Formulation

### 2.1. Dynamics of the Spherical LCE Balloon

Figure 1 depicts a spherical light-powered beating LCE balloon under periodic illumination. The light illumination is on/off with amplitude $I_{0}$ and zero. We assume that the azobenzene liquid-crystal molecules are planar anchored and randomly distributed in the LCE membrane of the balloon. According to Yu et al. [63], ultraviolet light or laser with wavelength less than 400 nm can induce the trans-to-cis isomerization of LCE. When the light shines on the surface of the balloon, a driving force is generated that causes the balloon to contract. When the light is off, the driving force gradually decreases, and the balloon gradually expands. The periodic light illumination may drive the periodic vibration of the LCE balloon. The LCE balloon without stress is taken as the reference state, in which the corresponding radius, thickness and mass density of LCE balloon are denoted by $r_{0}$, $h_{0}$, ρ, respectively. We suppose that the material of the LCE balloon is incompressible and its volume is always $V_{L}$. Then, the LCE balloon is filled with gas with amount of substance $n_{I}$, inducing the radius of the balloon increased to $r_{I}$, which is treated as the initial state. Afterwards, the LCE balloon is placed in the periodic light region, and its radius will vary with time under the combined action of the periodic light and the inner pressure of the balloon. LCE balloon will contract under illumination and recover under non-illumination conditions, and the instantaneous radius of the LCE balloon is described as r(t). Considering the incompressibility of the material, it is exactly in this sense that the thickness of the LCE balloon is regarded to be much smaller than the radius, and thus it can be defined as $h=V_{L}/4\pi r^{2}$. 

For simplicity, the influence of gravity is ignored. Since the sphere is completely symmetrical, we analyze the force condition per unit volume. A spherical shell volume element in the LCE balloon is selected; through force analysis, the element is subjected to the gas pressure $p_{\mathrm{in}}$ inside the balloon, the tensile stress σ in the LCE membrane of the balloon, the damping force and the total external pressure $p_{\mathrm{ext}}=p_{\mathrm{load}}+p_{\mathrm{am}}$, in which $p_{\mathrm{load}}$ indicates the loading pressure and $p_{\mathrm{am}}$ denotes the ambient pressure. Considering Newton’s laws of dynamics, the thermodynamic versus kinetic reaction control equation for the LCE element can be depicted as
(1)$$p_{\mathrm{in}}ds^{2}-p_{\mathrm{ext}}ds^{2}-4\sigma hds\cdot \frac{1}{2}\frac{ds}{r}-\beta \frac{dr}{dt}ds^{2}=\rho h\frac{d^{2}r}{dt^{2}}ds^{2}$$
where $4\sigma hds\cdot \frac{1}{2}\frac{ds}{r}$ is the force applied by the tensile stress in the LCE membrane of the LCE balloon as shown in Figure 1c, ds is the edge length of the spherical shell volume element of the LCE balloon, and β is the damping coefficient. With the assumption that gas inside the balloon is the ideal gas, the equation of state will be $p_{\mathrm{in}}V=n_{I}RT_{e}$, where R denotes the ideal gas constant, $T_{e}$ represents the thermodynamic temperature of the ideal gas, and $V=4\pi r^{3}/3$ is the gas volume. As stated above, the expression of the gas pressure inside the LCE balloon is obtained, which is $p_{\mathrm{in}}=3n_{I}RT_{e}/4\pi r^{3}$. Equation (1) can be reduced to
(2)$$p_{\mathrm{in}}-p_{\mathrm{ext}}-2\sigma \frac{h}{r}-\beta \frac{dr}{dt}=\rho h\frac{d^{2}r}{dt^{2}}$$

To focus on the oscillation of LCE balloon under periodic illumination, for simplicity, we assume that the stiffness of the LCE is constant during the oscillation, and the stress–strain relation is given as
(3)$$\sigma =E_{\mathrm{eff}}\varepsilon$$
with $E_{\mathrm{eff}}=2E/3$ indicating the effective Young’s modulus in the equiaxial stress state, and $\varepsilon =\left[r-r_{0}\left(1+\varepsilon_{L}\right)\right]/r_{0}\left(1+\varepsilon_{L}\right)$ denoting the elastic strain, where $\varepsilon_{L}$ is the effective light-induced contraction strain. To keep things simple, the effective light-induced contraction $\varepsilon_{L}$ is approximated as being proportional to the number fraction of cis-isomers φ(t), i.e.,
(4)$$\varepsilon_{L}=-C_{0}\varphi \left(t\right)$$
with $C_{0}$ being the contraction coefficient. Section 2.2 will provide a further description of the number fraction φ(t). 

### 2.2. Dynamic LCE Model

It is clearly known that after obtaining the number fraction of cis-isomers in the LCE balloon, the light-induced contraction can be calculated, and then the dynamics of the LCE balloon can be studied. Here, we provide a brief summary of the well-established dynamic LCE model. As illustrated in the research work of Yu et al. [63], ultraviolet light or laser with a wavelength less than 400 nm can induce the *trans*-to-*cis* isomerization of LCE. With the absorption of light energy, the rod-like *trans**-isomers* isomerize to the bent *cis-isomers*; meanwhile, due to the thermal effect, some of the *cis* isomers convert back to *trans* ones. Hence, the number fraction of *cis-isomers* is dependent on the light-driven *trans*-to-*cis* isomerization, the thermal excitation from *trans* to *cis*, and the thermal *cis*-to-*trans* relaxation. In general, the thermal excitation from *trans* to *cis* is often negligible compared to the light-driven *trans*-to-*cis* isomerization. The number fraction of *cis-isomers* bent in LCE can be described by the following rate equation [63]: (5)$$\frac{\partial \varphi }{\partial t}=\eta_{0}I_{0}(1-\varphi )-\frac{\varphi }{\tau }$$
with τ representing the thermal *cis*-to-*trans* relaxation time, $I_{0}$ indicating the light intensity, and $\eta_{0}$ denoting the light absorption constant. By solving Equation (5), the number fraction of *cis-isomers* can be given as
(6)$$\varphi (t)=\frac{\eta_{0}\tau I_{0}}{\eta_{0}\tau I_{0}+1}+(\varphi_{0}-\frac{\eta_{0}\tau I_{0}}{\eta_{0}\tau I_{0}+1})\exp\left[-\frac{t}{\tau }(\eta_{0}\tau I_{0}+1)\right]$$
where $\varphi_{0}$ is the number fraction of *cis-isomers* at the initial moment t=0 of a process. If the initial number fraction of the *cis-isomers* is zero in the illuminated state, that is, $\varphi_{0}=0$, Equation (6) can be simplified to
(7)$$\varphi (t)=\frac{\eta_{0}\tau I_{0}}{\eta_{0}\tau I_{0}+1}\left\{1-\exp\left[-\frac{t}{\tau }\left(1+\eta_{0}\tau I_{0}\right)\right]\right\}$$

In addition, for the non-illuminated state, namely $I_{0}=0$, Equation (6) can be modified into
(8)$$\varphi \left(t\right)=\varphi_{0}\exp\left(-\frac{t}{\tau }\right)$$
where the undetermined $\varphi_{0}$ can be set to the maximum value of φ(t) in Equation (7), which is $\varphi_{0}=\eta_{0}\tau I_{0}/\left(\eta_{0}\tau I_{0}+1\right)$. Therefore, the number fraction of *cis-isomers* in Equation (8) can be rewritten as
(9)$$\varphi \left(t\right)=\frac{\eta_{0}\tau I_{0}}{\eta_{0}\tau I_{0}+1}\exp\left(-\frac{t}{\tau }\right)$$

It is noted that Equations (7) and (9) provide the evolution laws of the number fraction in illuminated and non-illuminated states, and the current cis number fraction at the conversion moment between the two states does not change, as described in Section 2.4. 

### 2.3. Nondimensionalization

To nondimensionalize the above governing equation, we take the following dimensionless quantities into account: $\bar{t}=t/\tau$, $\bar{I}=\eta_{0}\tau I_{0}$, $\bar{r}=r/r_{0}$, ${\bar{r}}_{I}=r_{I}/r_{0}$, ${\bar{p}}_{L}=p_{L}/E_{\mathrm{eff}}$, ${\bar{p}}_{\mathrm{ext}}=p_{\mathrm{ext}}/E_{\mathrm{eff}}$, ${\bar{n}}_{I}=3n_{I}RT_{e}/4\pi E_{\mathrm{eff}}r_{0}^{3}$, ${\bar{V}}_{L}=V_{L}/4\pi r_{0}^{3}$, $\bar{\beta }=\beta r_{0}/E_{\mathrm{eff}}\tau$, $\bar{\rho }=\rho r_{0}^{2}/E_{\mathrm{eff}}\tau_{}^{2}$ and $\bar{\varphi }=\varphi \left(\eta_{0}\tau I_{0}+1\right)/\eta_{0}\tau I_{0}$.

In this paper, the illumination time and darkness time in one cycle are $T_{\mathrm{on}}$ and $T_{\mathrm{off}}$, respectively. The light illumination period is defined as $T_{L}=T_{\mathrm{on}}+T_{\mathrm{off}}$, and $T_{\mathrm{on}}/T_{L}$ denotes the illumination time rate. It is noted that $T_{\mathrm{on}}$ and $T_{\mathrm{off}}$ are independent of the thermal *cis*-to-*trans* relaxation time. Their corresponding dimensionless parameters are ${\bar{T}}_{\mathrm{on}}=T_{\mathrm{on}}/\tau$, ${\bar{T}}_{\mathrm{off}}=T_{\mathrm{off}}/\tau$, and ${\bar{T}}_{L}={\bar{T}}_{L}/\tau$. *A* is the amplitude of the oscillation of the balloon, and $\bar{\dot{r}}$ is the dimensionless velocity. 

In the illuminated state, the dimensionless expression of Equation (7) is given as
(10)$$\bar{\varphi }=1-\exp\left[-\bar{t}(\bar{I}+1)\right]$$

Additionally, for the non-illuminated state, the nondimensionalized Equation (9) can be expressed as
(11)$$\bar{\varphi }=\exp(-\bar{t})$$

By substituting Equations (3) and (4) into Equation (2), the dimensionless governing equation of the LCE balloon can be demonstrated as
(12)$$\frac{d^{2}\bar{r}}{d{\bar{t}}^{2}}=-\frac{\bar{\beta }{\bar{r}}^{2}}{\bar{\rho }{\bar{V}}_{L}}\frac{d\bar{r}}{d\bar{t}}+\frac{{\bar{n}}_{I}}{\bar{\rho }{\bar{V}}_{L}\bar{r}}-2\frac{\bar{r}-1+C_{0}\bar{\varphi }}{\bar{r}\bar{\rho }\left(1-C_{0}\bar{\varphi }\right)}-\frac{{\bar{p}}_{\mathrm{ext}}{\bar{r}}^{2}}{\bar{\rho }{\bar{V}}_{L}}$$
where $\bar{\varphi }$ is determined by the evolution laws of Equations (10) and (11), and its evolution is described in detail in Section 2.4. 

### 2.4. Solution Method

Equation (12) presents ordinary differential equations with variable coefficients for which there is no analytic solution. The ordinary differential equations are numerically solved in this paper using the classical fourth-order Runge–Kutta method using *MATLAB* software. The second-order ordinary differential equation with variable coefficients is transformed into two first-order ordinary differential equations with variable coefficients. Thus, the control equations can be rewritten as
(13)$$\{\begin{array}{c} \frac{d\bar{r}(\bar{t})}{d\bar{t}}=\bar{\dot{r}},\\ \frac{d^{2}\bar{r}}{{d\bar{t}}^{2}}=f(\bar{t},\bar{r},\bar{\dot{r}})\\ \bar{\dot{r}}(\bar{t}=0)={\bar{\dot{r}}}_{I},\\ \bar{r}(\bar{t}=0)={\bar{r}}_{I}. \end{array},$$

Consequently, the final steady-state responses of the balloon can be iteratively obtained. 

As Figure 2 shows, in an illumination cycle, when the LCE balloon switches between light-on and light-off states, the evolution law is correspondingly converted between Equations (10) and (11). When the LCE balloon is in the illumination state, the number of cis-isomers increases with the passage of time. When the balloon is in the non-illumination state, the number of cis-isomers decreases with the passage of time. It should be noted that at the moment of conversion between light-on and light-off states, the transient cis number fraction $\bar{\varphi }(t)$ keeps unchanged. For example, during the first illumination time ${\bar{T}}_{\mathrm{on}}$, $\bar{\varphi }\left(\bar{t}\right)$ rises from coordinate origin to B along the “light on” curve. Next, during the first darkness time ${\bar{T}}_{\mathrm{off}}$, $\bar{\varphi }\left(\bar{t}\right)$ falls from C to D along the “light off” curve. In the second illumination time ${\bar{T}}_{\mathrm{on}}$, $\bar{\varphi }\left(\bar{t}\right)$ transforms from point D to point A. 

## 3. Results and Discussion

In this section, we present a brief discussion on the beating of LCE balloons driven by periodic illumination. Firstly, the optimal illumination period and illumination time rate are numerically determined to maximize the vibration amplitude of the system. Then, under the optimum illumination conditions, the effects of light intensity, contraction coefficient, amount of gaseous substance, volume of LCE balloon, mass density, external pressure and damping coefficient on the vibration amplitude are further investigated. The beating amplitude of the LCE balloon can be adjusted by regulating the periodic lighting, material properties and geometric parameters. 

### 3.1. Dimensionless Parameters

To study the beating of the LCE balloon in detail, we need to estimate the typical values of the dimensionless parameters. From the accessible experiments [3,61,64,65], the material properties and geometric parameters of the system are shown in Table 1, and the dimensionless parameters are estimated as shown in Table 2. Thus, we can calculate the dimensionless light intensity, damping coefficient, amount of gaseous substances, etc. In the following, we explore the beating of balloons with different material properties under different periodic light patterns. 

### 3.2. Optimal Illumination Period 

Figure 3 shows the time course diagrams of the LCE balloon for different illumination periods, with the other parameters at ${\bar{T}}_{\mathrm{on}}/{\bar{T}}_{L}=0.7$, $C_{0}=0.3$, $\bar{I}=0.5$, ${\bar{n}}_{I}=0.45\left({\bar{r}}_{I}=1.164\right)$, ${\bar{V}}_{L}=0.05$, $\bar{\rho }=0.16$, ${\bar{p}}_{\mathrm{ext}}=0.18$, $\bar{\beta }=0.14$, and ${\bar{\dot{r}}}_{I}=0$. It can be observed from Figure 3 that after the initial non-periodic oscillation, the LCE balloon gradually stabilizes in the vibration. The beatings of the balloon are initially composed of transient vibrations related to free oscillations and steady-state vibrations relating to forced oscillations; then, the free oscillations disappear due to damping, and finally the balloon exhibits only the steady-state response of forced oscillations. It can be seen that the steady-state beating period is equal to the illumination period, and the illumination period has a great influence on the amplitude of the beating. With the different illumination periods, the balloon undergoes two different oscillation states, that is the single peak steady-state oscillation shown in Figure 3a,b, and the multimodal steady-state oscillation shown in Figure 3c,d. Multi-peak oscillation is relatively complex. Here, we only consider the case with a single-peak steady-state oscillation to keep things simple. 

Figure 4 presents the variation of the beating amplitude of the balloon with the illumination period at different light time rates. The other parameters in the calculation are set to $C_{0}=0.3$, $\bar{I}=0.5$, ${\bar{n}}_{I}=0.45\left({\bar{r}}_{I}=1.164\right)$, ${\bar{V}}_{L}=0.05$, $\bar{\rho }=0.16$, ${\bar{p}}_{\mathrm{ext}}=0.18$, $\bar{\beta }=0.14$, and ${\bar{\dot{r}}}_{I}=0$. As can be clearly seen from the figure, the amplitude first increases to the maximum value, and then decreases with increasing illumination period. The LCE balloon beating amplitude reaches its maximum when the illumination period is around 2.7. It is important to note that the optimal illumination period does not vary with the light time rate. 

### 3.3. Optimal Illumination Time Rate

The variation of the beating amplitude of the balloon with the optimal illumination time rate under the optimal illumination period is plotted in Figure 5. The other parameters are ${\bar{T}}_{L}=2.7$, $C_{0}=0.3$, $\bar{I}=0.5$, ${\bar{n}}_{I}=0.45\left({\bar{r}}_{I}=1.164\right)$, ${\bar{V}}_{L}=0.05$, $\bar{\rho }=0.16$, ${\bar{p}}_{\mathrm{ext}}=0.18$, $\bar{\beta }=0.14$, and ${\bar{\dot{r}}}_{I}=0$. With increasing illumination time rate, the beating amplitude first increases from zero to its maximum value, and then decreases back to zero. The maximum amplitude corresponds to the illumination time rate ${\bar{T}}_{\mathrm{on}}/{\bar{T}}_{L}=0.5$. The beating amplitude is symmetrical around the time rate of 0.5, and the beating amplitude on the axis of symmetry is the maximum. It is easy to draw the conclusion that by adjusting the illumination time rate, we can control the beating amplitude in soft robot driving and energy acquisition systems. 

### 3.4. Effect of Light Intensity

Figure 6 shows the effect of light intensity $\bar{I}$ on the beating of the LCE balloon under the optimal illumination period and optimal time rate, with the other parameters set to ${\bar{T}}_{\mathrm{on}}/{\bar{T}}_{L}=0.5$, ${\bar{T}}_{L}=2.7$, $C_{0}=0.3$, ${\bar{n}}_{I}=0.45\left({\bar{r}}_{I}=1.164\right)$, ${\bar{V}}_{L}=0.05$, $\bar{\rho }=0.16$, ${\bar{p}}_{\mathrm{ext}}=0.18$, $\bar{\beta }=0.14$, and ${\bar{\dot{r}}}_{I}=0$. The time history curves and limit cycles of beating of the balloon under three different light intensities are plotted in Figure 6a,b respectively. As can be seen from the figure, with increasing light intensity, the beating amplitude increases, but the beating period remains unchanged, always being equivalent to the illumination period. The results indicate that the light intensity affects the amplitude of steady-state beating. This is because the light-induced contraction strain of the LCE balloon becomes larger with increasing light intensity, and therefore the amplitude increases. 

### 3.5. Effect of Contraction Coefficient

Figure 7 reveals the influence of contraction coefficient $C_{0}$ on the beating of the LCE balloon under the optimal illumination period and optimal time rate, with the other parameters set to ${\bar{T}}_{\mathrm{on}}/{\bar{T}}_{L}=0.5$, ${\bar{T}}_{L}=2.7$, $\bar{I}=0.5$, ${\bar{n}}_{I}=0.45\left({\bar{r}}_{I}=1.164\right)$, ${\bar{V}}_{L}=0.05$, $\bar{\rho }=0.16$, ${\bar{p}}_{\mathrm{ext}}=0.18$, $\bar{\beta }=0.14$, and ${\bar{\dot{r}}}_{I}=0$. Figure 7a plots the time history of the balloon beating under three different contraction coefficients. Figure 7b shows the limit cycles of the LCE balloon under three different contraction coefficients. It can be observed that with increasing contraction coefficient, the beating amplitude shows an increasing trend, but the beating period remains at the same value as the illumination period. The results prove that the contraction coefficient has an influence on the amplitude of steady-state oscillation. This is because with larger values of contraction coefficient, the magnitude of light-induced contraction strain is larger, and thus the amplitude increases. For different contraction coefficients, the LCE balloon can complete beating under the switching between illumination state and non-illumination state. 

### 3.6. Effect of Amount of Substance

Figure 8 depicts the effect of amount of gaseous substance ${\bar{n}}_{I}$ on beating under the optimal illumination period and optimal time rate, with the other parameters set to ${\bar{T}}_{\mathrm{on}}/{\bar{T}}_{L}=0.5$, ${\bar{T}}_{L}=2.7$, $\bar{I}=0.5$, $C_{0}=0.3$, ${\bar{V}}_{L}=0.05$, $\bar{\rho }=0.16$, ${\bar{p}}_{\mathrm{ext}}=0.18$, $\bar{\beta }=0.14$, and ${\bar{\dot{r}}}_{I}=0$. Figure 8a plots the evolution of the balloon subjected to beatings for different amount of gaseous substance. Figure 8b plots the limit cycles of the LCE balloon for three different amounts of gaseous substance. As can be seen from the graph, the amplitude of the beating increases with increasing amount of gaseous substance; however, the period of beating remains the same as the illumination period. It is indicated that the amplitude of the steady-state oscillations is affected by the amount of gaseous substance. For different amounts of gaseous substance, the LCE balloon can accomplish beating through switching between illuminated and non-illuminated states. 

### 3.7. Effect of LCE Volume

The influence of volume ${\bar{V}}_{L}$ on the beating of the balloon at the optimal illumination period and the optimal time rate, is presented in Figure 9, with the other parameters set to ${\bar{T}}_{\mathrm{on}}/{\bar{T}}_{L}=0.5$, ${\bar{T}}_{L}=2.7$, $\bar{I}=0.5$, $C_{0}=0.3$, ${\bar{n}}_{I}=0.45\left({\bar{r}}_{I}=1.164\right)$, $\bar{\rho }=0.16$, ${\bar{p}}_{\mathrm{ext}}=0.18$, $\bar{\beta }=0.14$, and ${\bar{\dot{r}}}_{I}=0$. Figure 9a shows the time history of the beating of balloons under different volumes. Figure 9b shows the limit cycles of the LCE balloon under different volumes. As shown in the figure, the LCE balloon presents a beating mode. With increasing balloon volume, the beating amplitude increases; in addition, the beating period remains equal to the illumination period. It can be concluded that the balloon volume has an effect on the amplitude of steady-state oscillation; moreover, for different volumes, the LCE balloon can complete beating under switching between illumination state and non-illumination state. 

### 3.8. Effect of Mass Density

With the other parameters at ${\bar{T}}_{\mathrm{on}}/{\bar{T}}_{L}=0.5$, ${\bar{T}}_{L}=2.7$, $\bar{I}=0.5$, $C_{0}=0.3$, ${\bar{n}}_{I}=0.45\left({\bar{r}}_{I}=1.164\right)$, ${\bar{p}}_{\mathrm{ext}}=0.18$, ${\bar{V}}_{L}=0.05$, $\bar{\beta }=0.14$, and ${\bar{\dot{r}}}_{I}=0$, Figure 10 presents the effect of mass density $\bar{\rho }$ on beating at the optimal illumination period and optimal time rate. The amplitude evolution and limit cycles of the beatings of the balloon under different mass densities are presented in Figure 10a,b, respectively. It can be seen from the figure that the LCE balloon is in the beating mode. With increasing balloon mass density, the beating amplitude increases, but the beating period always remains equivalent to the illumination period. The mass density is proved to be capable of affecting the amplitude of steady-state oscillation. For different mass densities, the LCE balloon can realize beating under the transition between illumination and non-illumination. 

### 3.9. Effect of External Pressure

As shown in Figure 11, the influence of the external pressure ${\bar{p}}_{0}$ on beating is illustrated under the optimal illumination period and optimal time rate, with the other parameters set to ${\bar{T}}_{\mathrm{on}}/{\bar{T}}_{L}=0.5$, ${\bar{T}}_{L}=2.7$, $\bar{I}=0.5$, $C_{0}=0.3$, ${\bar{n}}_{I}=0.45\left({\bar{r}}_{I}=1.164\right)$, ${\bar{V}}_{L}=0.05$, $\bar{\rho }=0.16$, $\bar{\beta }=0.14$, and ${\bar{\dot{r}}}_{I}=0$. Figure 11a shows the amplitude evolution of beating of LCE balloon under different external pressures. Figure 11b shows the limit cycles of the LCE balloon under three different external pressures. It can be clearly seen from the figure that the LCE balloon experiences a beating motion. With increasing external pressure, the beating amplitude undergoes a slight decrease; however, the beating period does not change, and it remains equal to the illumination period. The results reveal that the external pressure will have an influence on the amplitude of steady-state oscillation. For different values of external pressure, the beating of the LCE balloon can be completed under switching between the illumination state and non-illumination state. 

### 3.10. Effect of Damping Coefficient

Figure 12 provides the effect of damping coefficient $\bar{\beta }$ on beating under the optimal illumination period and optimal time rate, with the other parameters set to ${\bar{T}}_{\mathrm{on}}/{\bar{T}}_{L}=0.5$, ${\bar{T}}_{L}=2.7$, $\bar{I}=0.5$, $C_{0}=0.3$, ${\bar{n}}_{I}=0.45\left({\bar{r}}_{I}=1.164\right)$, ${\bar{V}}_{L}=0.05$, $\bar{\rho }=0.16$, ${\bar{p}}_{\mathrm{ext}}=0.18$, and ${\bar{\dot{r}}}_{I}=0$. Figure 12a,b show the time history diagrams and limit cycles of beating of balloon under different damping coefficients, respectively. It can be clearly observed from the figure that the LCE balloon is in the beating mode. With increasing damping coefficient, the beating amplitude decreases; additionally, the beating period remains the same, always being equivalent to the illumination period. Obviously, the damping coefficient has an effect on the amplitude of steady-state oscillation. This is due to the energy competition between energy input from illumination and damping-induced energy dissipation. With increasing damping coefficient, the energy dissipation induced by damping increases, resulting in a decrease in beating amplitude. 

### 3.11. Effect of Initial Velocity

Figure 13 indicates the effect of initial velocity ${\bar{\dot{r}}}_{I}$ on the beating of the balloon under the optimal illumination period and the optimal time rate, with the other parameters set to ${\bar{T}}_{\mathrm{on}}/{\bar{T}}_{L}=0.5$, ${\bar{T}}_{L}=2.7$, $\bar{I}=0.5$, $C_{0}=0.3$, ${\bar{n}}_{I}=0.45\left({\bar{r}}_{I}=1.164\right)$, ${\bar{V}}_{L}=0.05$, $\bar{\rho }=0.16$, ${\bar{p}}_{\mathrm{ext}}=0.18$, and $\bar{\beta }=0.14$. Figure 13a shows the evolution of the beating of the balloon at different initial velocities under the optimal illumination period and optimal time rate. Figure 13b shows the limit cycles of the LCE balloon at three different initial velocities. It can be clearly seen that the LCE balloon is in the beating mode, and the beating amplitude does not change with increasing initial velocity. Meanwhile, the beating period remains equal to the illumination period. The results show that the initial velocity has no influence on the amplitude of beating, which is consistent with the general characteristics of forced oscillation. 

### 3.12. An Application Example of the Periodic Oscillation of the Balloon

The oscillating system proposed in this paper has the potential to be developed as a light-fueled micro-pump. The LCE balloon could be placed in a sealed transparent glass sphere with a check valve at each end. When the LCE balloon expands, the water in the glass sphere is squeezed out from the outlet check valve, and when the LCE balloon contracts, the water flows into the glass sphere from the inlet check valve. In practical applications, the energy/power density and energy conversion efficiency highly depend on the specific energy conversion processes. For the simple model developed in the current study, during the periodic oscillation of the LCE balloon, the light energy absorbed by the system is used to compensate for the damping energy and to do work on connected devices. For this light-driven micro-pump, the work done by the system on the connected devices can be considered to be the effective work of the pump. 

For the typical values of $I=2.5\;{W/\mathrm{cm}}^{2}$, $\rho =10^{3}\;{\mathrm{kg}/m}^{3}$, $r_{I}=1.164\;\mathrm{mm}$, τ=0.1 s, β=8.02 s⋅MPa/m, E=1 MPa, $\eta_{0}=0.00022\;m^{2}/s\cdot w$, $C_{0}=0.3$, and the mass density of the fiber $\rho_{f}=10^{3}\;{\mathrm{kg}/m}^{3}$, the dimensionless parameters are calculated as ${\bar{T}}_{\mathrm{on}}/{\bar{T}}_{L}=0.5$, $\bar{I}=0.55$, $C_{0}=0.3$, ${\bar{n}}_{I}=0.45\left({\bar{r}}_{I}=1.164\right)$, ${\bar{V}}_{L}=0.05$, $\bar{\rho }=0.16$, ${\bar{p}}_{\mathrm{ext}}=0.18$, $\bar{\beta }=0.14$ and ${\bar{\dot{r}}}_{I}=0$. In this case, the maximum and minimum values of the balloon radius during beating are numerically calculated to be ${\bar{r}}_{2}=1.164$ and ${\bar{r}}_{1}=1.145$, respectively, and thus the effective work on the external connected equipment during the expansion of the balloon in a period is calculated as $W=E_{\mathrm{eff}}{\bar{p}}_{\mathrm{ext}}\frac{4}{3}\pi r_{0}^{3}\left({\bar{r}}_{2}^{3}-{\bar{r}}_{1}^{3}\right)=3.82\times 10^{-5}J$. In addition, the dimensionless period can be numerically calculated as $\bar{T}=2.69$. By inserting $T_{\mathrm{on}}={\bar{T}}_{\mathrm{on}}\tau =0.1345\;s$, the average power of the external work done by the pump is $P=W/T_{\mathrm{on}}=2.84\times 10^{-4}\;W$. 

## 4. Conclusions

The light-driven oscillation of LCE can convert light energy into mechanical energy, which is of great significance for soft robots, microfactories, and nanomachines. In this paper, the active control of spherical LCE balloon beating driven by periodic illumination is discussed theoretically. Based on dynamic LCE model and Newton’s laws of dynamics, the governing equations are derived, and the numerical solution method of the dynamic equations are given. The numerical results validate that LCE balloon can beat periodically under periodic illumination, and the beating period is related to the period of illumination. For the maximum steady-state amplitude of periodic beating, an optimal periodic illumination condition is exhibited. With increasing illumination time rate, the beating amplitude first increases, reaches its maximum, and then decreases. The amplitude of beating can be accurately controlled by adjusting the light intensity, illumination period, illumination time rate, etc. The beating LCE balloon in this study has promising application prospects in the fields of opto-mechanical energy conversion systems and light-fueled machines and equipment, such as cardiac pacemakers, advanced robotics, and so on. 

## Figures and Tables

**Figure 1 micromachines-13-00769-f001:**
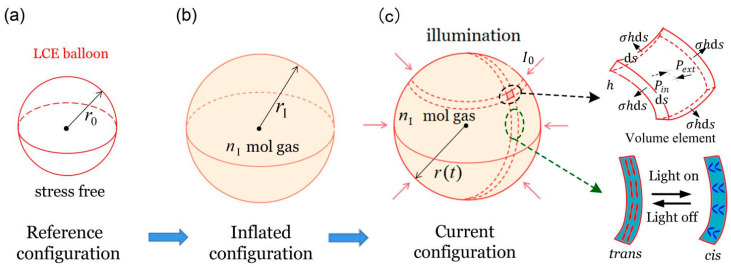
Schematic diagram of a spherical light-powered beating LCE balloon. (**a**) Reference state in stress-free state. (**b**) The balloon is inflated by gas with amount of substance $n_{I}$ to the balanced state, namely the initial state. (**c**) The instant state of the balloon as a function of the radius r(t) under periodic lighting conditions.

**Figure 2 micromachines-13-00769-f002:**
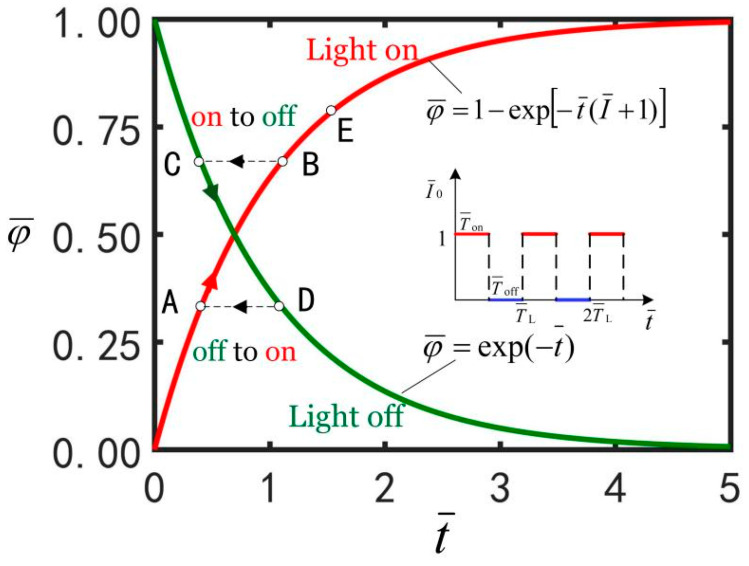
The evolution laws of the number fraction of cis isomers $\bar{\varphi }\left(\bar{t}\right)$ in the balloon with time $\bar{t}$.

**Figure 3 micromachines-13-00769-f003:**
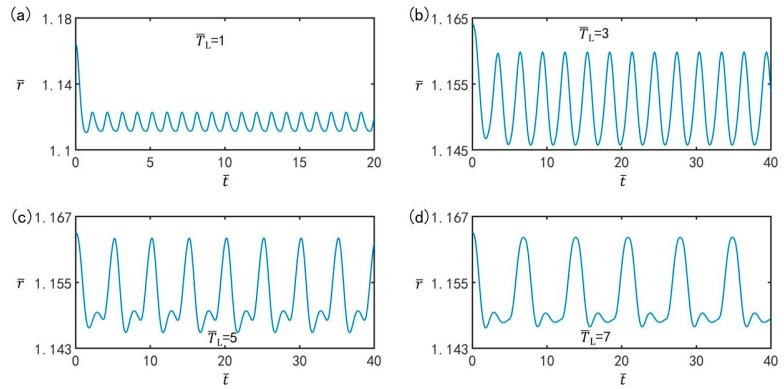
Time course diagrams of the LCE balloon forced to beat for period illumination (**a**) ${\bar{T}}_{L}=1$, (**b**) ${\bar{T}}_{L}=3$, (**c**) ${\bar{T}}_{L}=5$, and (**d**) ${\bar{T}}_{L}=7$. The parameters are ${\bar{T}}_{\mathrm{on}}/{\bar{T}}_{L}=0.7$, $C_{0}=0.3$, $\bar{I}=0.5$, ${\bar{n}}_{I}=0.45\left({\bar{r}}_{I}=1.164\right)$, ${\bar{V}}_{L}=0.05$, $\bar{\rho }=0.16$, ${\bar{p}}_{\mathrm{ext}}=0.18$, $\bar{\beta }=0.14$, and ${\bar{\dot{r}}}_{I}=0$. The illumination period affects the beating amplitude of the balloon.

**Figure 4 micromachines-13-00769-f004:**
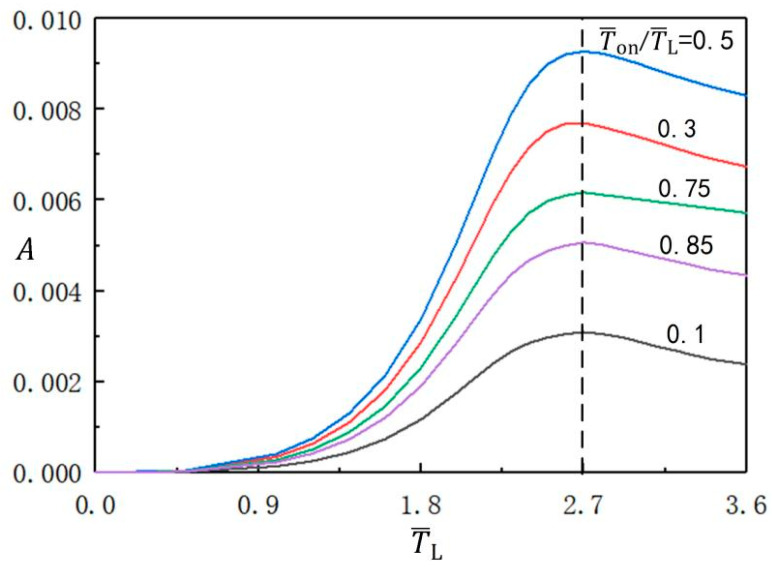
The influence of illumination period on the amplitude of the beating balloon, for different illumination time rates. The parameters are $C_{0}=0.3$, $\bar{I}=0.5$, ${\bar{n}}_{I}=0.45\left({\bar{r}}_{I}=1.164\right)$, ${\bar{V}}_{L}=0.05$, $\bar{\rho }=0.16$, ${\bar{p}}_{\mathrm{ext}}=0.18$, $\bar{\beta }=0.14$, and ${\bar{\dot{r}}}_{I}=0$. The amplitude first increases and then decreases with increasing illumination period.

**Figure 5 micromachines-13-00769-f005:**
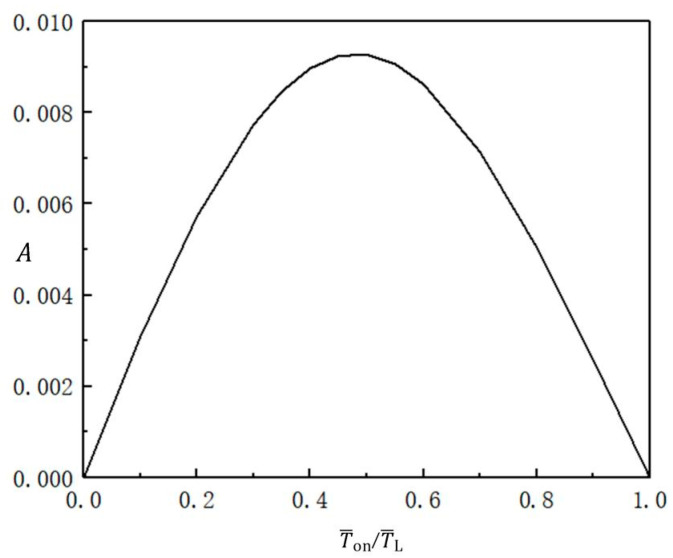
Variation of the beating amplitude of the balloon with the illumination time rate under the optimal illumination period. The parameters are ${\bar{T}}_{L}=2.7$, $C_{0}=0.3$, $\bar{I}=0.5$, ${\bar{n}}_{I}=0.45\left({\bar{r}}_{I}=1.164\right)$, ${\bar{V}}_{L}=0.05$, $\bar{\rho }=0.16$, ${\bar{p}}_{\mathrm{ext}}=0.18$, $\bar{\beta }=0.14$, and ${\bar{\dot{r}}}_{I}=0$. The amplitude first increases and then decreases with increasing illumination time rate.

**Figure 6 micromachines-13-00769-f006:**
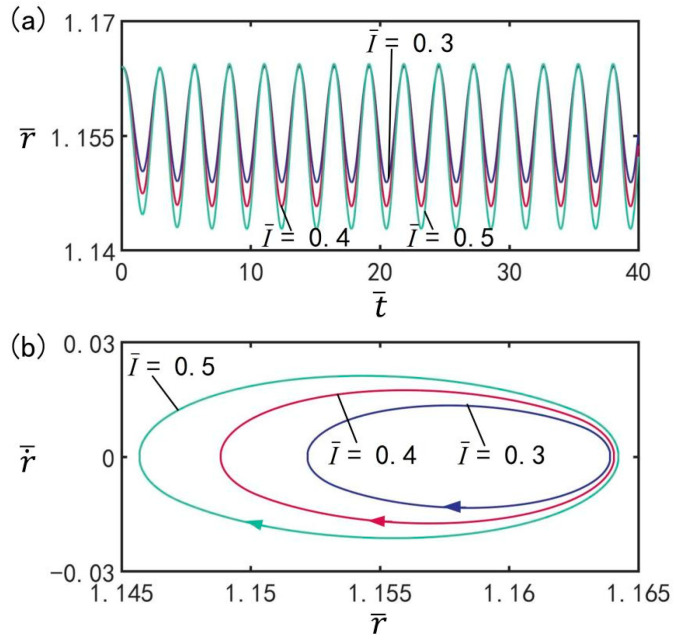
The effect of light intensity $\bar{I}$ on the light-powered beating of LCE balloon. (**a**) Time histories; (**b**) Limit cycles. The parameters are ${\bar{T}}_{\mathrm{on}}/{\bar{T}}_{L}=0.5$, ${\bar{T}}_{L}=2.7$, $C_{0}=0.3$, ${\bar{n}}_{I}=0.45\left({\bar{r}}_{I}=1.164\right)$, ${\bar{V}}_{L}=0.05$, $\bar{\rho }=0.16$, ${\bar{p}}_{\mathrm{ext}}=0.18$, $\bar{\beta }=0.14$, and ${\bar{\dot{r}}}_{I}=0$. With increasing light intensity, the beating amplitude increases.

**Figure 7 micromachines-13-00769-f007:**
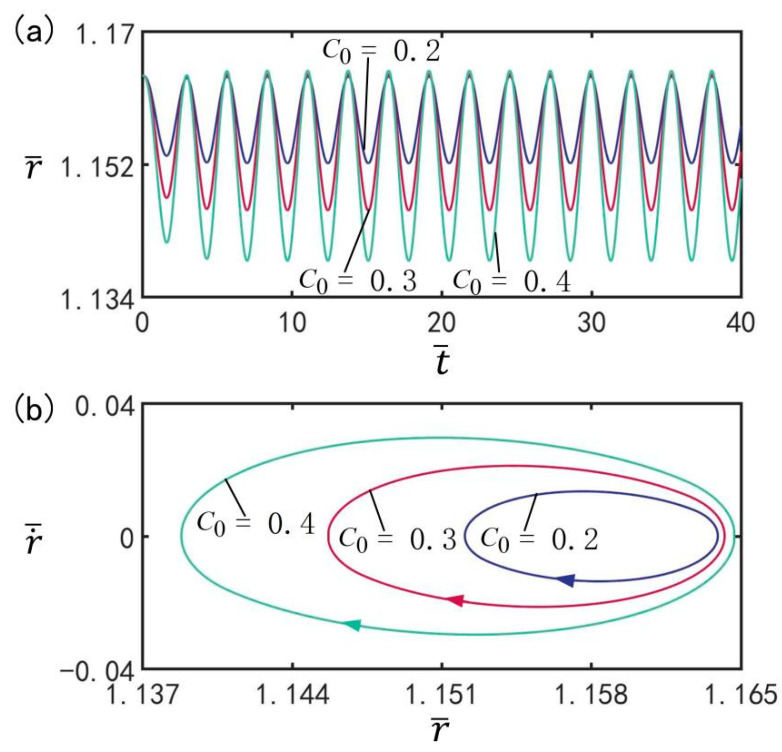
The influence of contraction coefficient $C_{0}$ on the beating of the LCE balloon. (**a**) Time histories; (**b**) Limit cycles. The parameters are ${\bar{T}}_{\mathrm{on}}/{\bar{T}}_{L}=0.5$, ${\bar{T}}_{L}=2.7$, $\bar{I}=0.5$, ${\bar{n}}_{I}=0.45\left({\bar{r}}_{I}=1.164\right)$, ${\bar{V}}_{L}=0.05$, $\bar{\rho }=0.16$, ${\bar{p}}_{\mathrm{ext}}=0.18$, $\bar{\beta }=0.14$, and ${\bar{\dot{r}}}_{I}=0$. With increasing contraction coefficient, the beating amplitude increases.

**Figure 8 micromachines-13-00769-f008:**
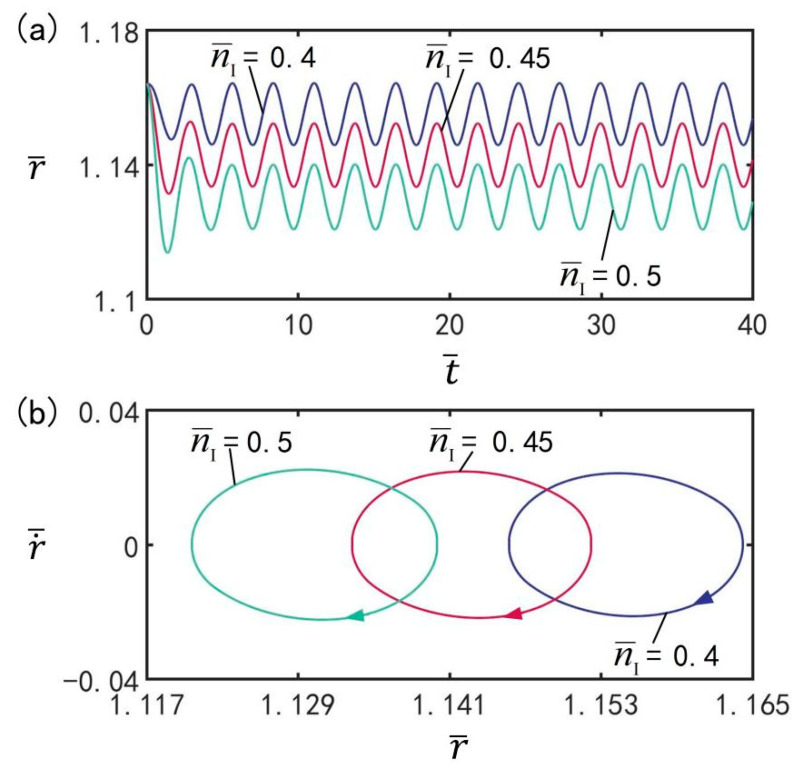
The effect of amount of gaseous substance ${\bar{n}}_{I}$ on the beating of the LCE balloon. (**a**) Time histories; (**b**) Limt cycles. The parameters are ${\bar{T}}_{\mathrm{on}}/{\bar{T}}_{L}=0.5$, ${\bar{T}}_{L}=2.7$, $\bar{I}=0.5$, $C_{0}=0.3$, ${\bar{V}}_{L}=0.05$, $\bar{\rho }=0.16$, ${\bar{p}}_{\mathrm{ext}}=0.18$, $\bar{\beta }=0.14$, and ${\bar{\dot{r}}}_{I}=0$. The amplitude of oscillations increases with increasing amount of gaseous substance.

**Figure 9 micromachines-13-00769-f009:**
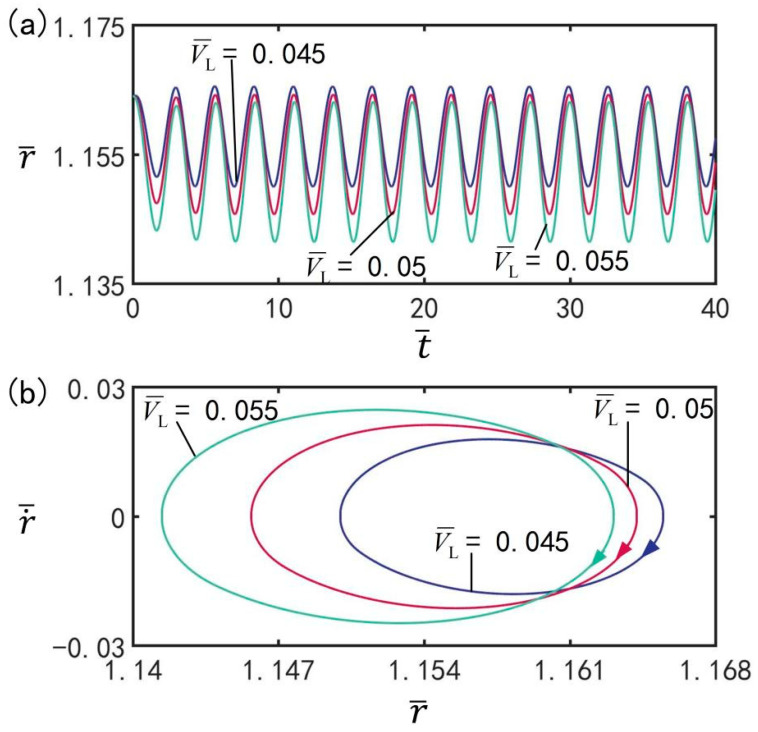
The effect of LCE balloon volume ${\bar{V}}_{L}$ on the beating of the LCE balloon. (**a**) Time histories; (**b**) Limit cycles. The parameters are ${\bar{T}}_{\mathrm{on}}/{\bar{T}}_{L}=0.5$, ${\bar{T}}_{L}=2.7$, $\bar{I}=0.5$, $C_{0}=0.3$, ${\bar{n}}_{I}=0.45\left({\bar{r}}_{I}=1.164\right)$, $\bar{\rho }=0.16$, ${\bar{p}}_{\mathrm{ext}}=0.18$, $\bar{\beta }=0.14$, and ${\bar{\dot{r}}}_{I}=0$. The beating amplitude increases with increasing balloon volume.

**Figure 10 micromachines-13-00769-f010:**
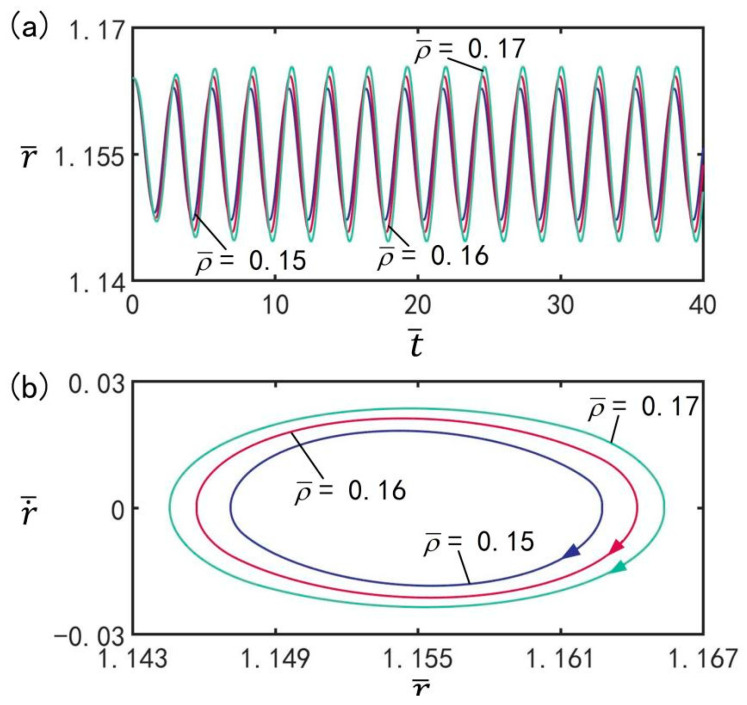
The effect of mass density $\bar{\rho }$ on the beating of LCE balloon. (**a**) Time histories; (**b**) Limit cycles. The parameters are ${\bar{T}}_{\mathrm{on}}/{\bar{T}}_{L}=0.5$, ${\bar{T}}_{L}=2.7$, $\bar{I}=0.5$, $C_{0}=0.3$, ${\bar{n}}_{I}=0.45\left({\bar{r}}_{I}=1.164\right)$, ${\bar{p}}_{\mathrm{ext}}=0.18$, ${\bar{V}}_{L}=0.05$, $\bar{\beta }=0.14$, and ${\bar{\dot{r}}}_{I}=0$. With increasing balloon mass density, the beating amplitude increases.

**Figure 11 micromachines-13-00769-f011:**
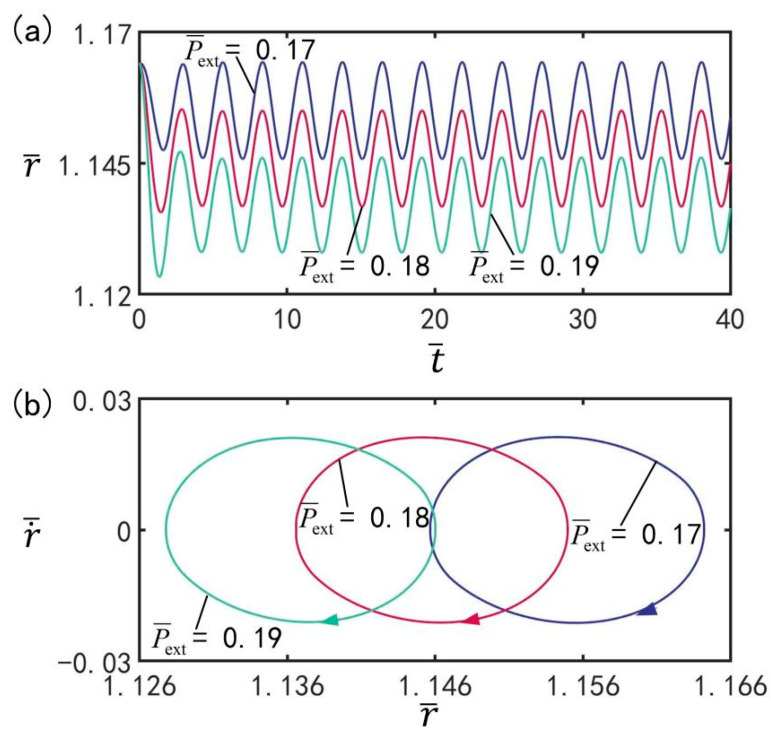
The influence of external pressure ${\bar{p}}_{0}$ on the beating of LCE balloon. (**a**) Time histories; (**b**) Limit cycles. The parameters are ${\bar{T}}_{\mathrm{on}}/{\bar{T}}_{L}=0.5$, ${\bar{T}}_{L}=2.7$, $\bar{I}=0.5$, $C_{0}=0.3$, ${\bar{n}}_{I}=0.45\left({\bar{r}}_{I}=1.164\right)$, ${\bar{V}}_{L}=0.05$, $\bar{\rho }=0.16$, $\bar{\beta }=0.14$, and ${\bar{\dot{r}}}_{I}=0$. With increasing external pressure, the beating amplitude decreases.

**Figure 12 micromachines-13-00769-f012:**
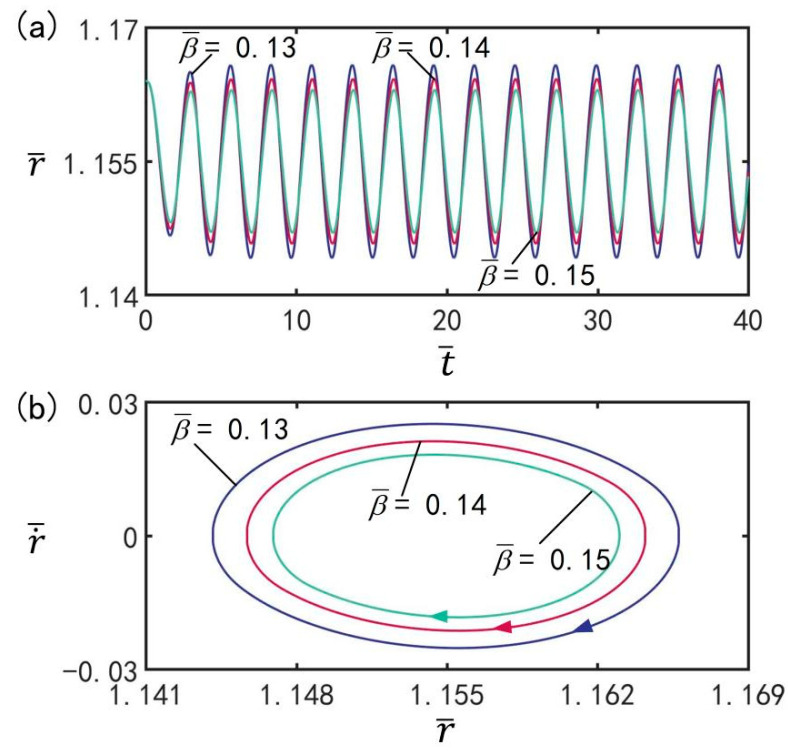
The effect of damping coefficient $\bar{\beta }$ on the beating of LCE balloon. (**a**) Time histories; (**b**) Limit cycles. The parameters are ${\bar{T}}_{\mathrm{on}}/{\bar{T}}_{L}=0.5$, ${\bar{T}}_{L}=2.7$, $\bar{I}=0.5$, $C_{0}=0.3$, ${\bar{n}}_{I}=0.45\left({\bar{r}}_{I}=1.164\right)$, ${\bar{V}}_{L}=0.05$, $\bar{\rho }=0.16$, ${\bar{p}}_{\mathrm{ext}}=0.18$, and ${\bar{\dot{r}}}_{I}=0$. With increasing damping coefficient, the beating amplitude decreases.

**Figure 13 micromachines-13-00769-f013:**
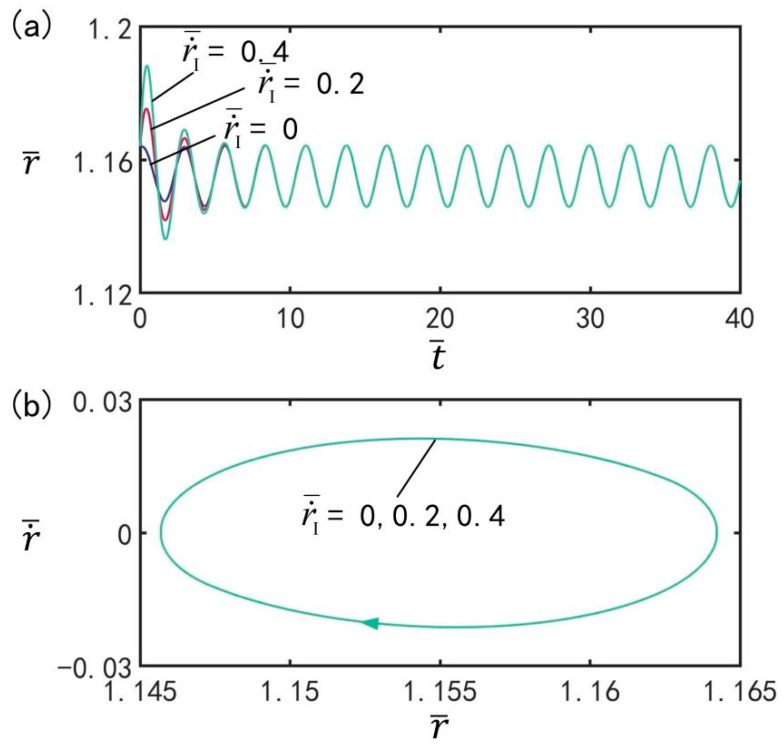
The effect of initial velocity ${\bar{\dot{r}}}_{I}$ on the beating of LCE balloon. (**a**) Time histories; (**b**) Limit cycles. The parameters are ${\bar{T}}_{\mathrm{on}}/{\bar{T}}_{L}=0.5$, ${\bar{T}}_{L}=2.7$, $\bar{I}=0.5$, $C_{0}=0.3$, ${\bar{n}}_{I}=0.45\left({\bar{r}}_{I}=1.164\right)$, ${\bar{V}}_{L}=0.05$, $\bar{\rho }=0.16$, ${\bar{p}}_{\mathrm{ext}}=0.18$, and $\bar{\beta }=0.14$. The initial velocity does not affect the beating amplitude.

**Table 1 micromachines-13-00769-t001:** Material properties and geometric parameters.

Parameter	Definition	Value	Units
τ	Thermal relaxation time	0.1	s
ρ	Mass density	1000~1200	${\mathrm{kg}/m}^{3}$
$I_{0}$	Light intensity	15~35	${\mathrm{kW}/m}^{2}$
β	Damping coefficient	8~10	s·MPa/m
$\eta_{0}$	Light-absorption constant	0.00022	$m^{2}/s\cdot W$
E	Young’s modulus	1	MPa
$r_{I}$	Initial radius	0.001164	m
$C_{0}$	Contraction coefficient	0.3	/

**Table 2 micromachines-13-00769-t002:** Dimensionless parameters.

$\bar{I}$	${\bar{n}}_{I}$	${\bar{V}}_{L}$	$\bar{\rho }$	${\bar{p}}_{\mathrm{ext}}$	$\bar{\beta }$	${\bar{\dot{r}}}_{I}$
0.3~0.5	0.4~0.5	0.045~0.055	0.15~0.18	0.17~0.19	0.12~0.15	0~0.4

## Data Availability

Not applicable.

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
