# Peer review of "Beating of a Spherical Liquid Crystal Elastomer Balloon under Periodic Illumination"

_micromachines, 2022, doi:10.3390/mi13050769_

Round 1

Reviewer 1 Report

see pdf document

Reviewer 2 Report

In this manuscript, Cheng et al. investigated the oscillation in a spherical LCE balloon. A dynamic response model was develoepd and its dynamic responses were analyzed. The effects of frequency, light intensity and many other parameters were considered. Overall, this paper is well written and the results are clearly presented. The research is comprehensive. The reviewer can recommend for publication with the following minor comments.

  1. The LCE stiffness is assumed to be a constant in this paper (Eq.2), is this linear approximation commonly adopted for LCE? Because sometimes we would use hyperelastic models for elastomers. Please explain.
  2. What are the potential applications of this study. I know that the authors mentioned some very general applications in the conclusion, but it would be better if the authors can be more specific (provide some specific examples where this study can be used) so that the readers can see that this paper can have clear impact on the LCEs.

Round 2

Reviewer 1 Report

Report on
"Beating of a spherical LCE balloon under periodic illumination"
The manuscript has been substantially improved. The authors have revised several passages and resolved most of the critical issues raised in my first report. In my view, the manuscript is now essentially free of obvious mistakes. 

The authors also reacted to the remark regarding the geometric üparameters, by choosing a much smaller ratio of balloon volume and elastomer volume, now 10 % of the previous value. They also made some estimate of the powers and efficiencies of the device.

I am still very skeptical that this device and the geometric parameters given in the manuscript have any practical relevance. Now, the elastomer layer thickness is only about 8 mm, but the authors should be aware that the typical elastomer sheets (which need to have monodomain orientation with the director everywhere orthogonal to the membrane surface, otherwise the contraction mechanism fails) are of the order of micrometers, hardly reaching one millimeter. I guess that it is hardly possible to create such thick ordered elastomer sheets. Also, the absorption of such a thick layer is not well explained. If the absorption coefficient is too small, then most of the light will pass the balloon and the efficiency will be low. If the absorption is strong, the inner layers of the membrane will not receive enough light intensity, and will not perform the trans-cis isomerization. Then the balloon will wrinkle.

The definition of the overlined I at page 5, line 143, is η0 τ I0. When I insert the values from Table 1, 0.00022 m2/Ws, 0.1 s and 20 kW/m2, I obtain 0.44. I have no clue how the authors compute with these  parameters their value of 0.25 that is given, e.g. in the caption of Figure 3. Also, it is not clear why the authors give a range of 0.1 to 0.5 for the same quantity in Table 2, when Table 1 already contains all parameters needed to compute the exact value. 

I have no idea what the factor T0 in the exponent of Eq. (6) means. In my opinion, it should be τ.

Equations (7) and (9) are correct only if the illumination and dark phases are long compared to the relaxation time τ. Figure (2) shows that this is fulfilled only when Ton and Toff are approximately 4 τ or longer.

The unit Watt (light absorption constant) should be abbreviated by an upper case W in Tab. 1.

In Figure 5, the scaled elastomer volume is again 1, in contrast to the more realistic value of 0.1 given above. It is probably a misprint.

These errors can probably be corrected, but I still doubt that the calculations describe a realistic system, not to speak of a micromachine, and I am very skeptical that the graphs of the velocity vs. radius in Figs. 5 to 12 are of sufficient interest for the general auditory. This is a purely academic calculation on undergraduate level.

Most problematic for me is that I cannot follow the calculation of the work done by the pump in the response letter. I use the classical equation for isothermal compression of an ideal gas, n1 RT ln (V1/V0). The logarithm of the volume ratio os roughly 0.5 (assuming a 60 % compression, estimated from the radius changes in the figures). The leading factor (n1 R T), where n1 is the amount of gas, is roughly equal to p V, and is 420 J. Thus the work needed to compress the gas is about 210 J, and not 23 J as assumed by the
authors. I do not know where this discrepancy comes from, but it should be settled before the manuscript can be published. Otherwise, the light intensity needed to perform this work would be an order of magnitude larger than assumed by the authors. 

Round 3

Reviewer 1 Report

Report on
"Beating of a spherical LCE balloon under periodic illumination"

The manuscript has been substantially improved. The authors have further revised several passages and resolved some critical issues raised in my second report.  

Respectfully, I disagree with two of the arguments given by the authors in their second response letter.

The first issue is of minor importance. I mentioned that Equations (7) and (9) are correct only if the illumination and dark phases are long compared to the relaxation time τ, i.e. when T_on and T_off are approximately 4 τ or longer. The authors claim that this is not the case and that the equations are correct for all periodicities.

In fact, this is not true in my opinion. Let us take Equation (9). At time t = 0 the variable φ has a value of η0 τ I0/(η0 τ I0 + 1). This is the value that φ adopts after long enough illumination periods, i.e. when the exponential in Eq. (7) has dropped to zero. If the illumination period was shorter, then this asymptotic value was not reached yet and the prefactor of the exponential in Eq. (9). On the other hand, the initial value of φ in Eq. (7) is zero. This is the case after a suficiently long dark period. Otherwise, the prefactors in that equation need to be corrected. That's why I mentioned that these equations are not valid for short period cyclic illumination. Under a very short period oscillation, a slightly oscillating mean φ would be expected. Probably, the authors mistook my comment.

The second issue is much more serious. 
The authors write "The classical equation for isothermal compression of an ideal gas is a calculation of the work done by the inner gas of the balloon, while this paper calculates the work done to the outside by the expanding 
balloon, which is regarded as the effective work."
Well, in my understanding the classical equation for isothermal (or probably adiabatic) volume changes of a gas describe both the work done by the gas when it expands from volume 1 to volume 2, but also the work that is needed for the compression of the same gas from a larger volume 2 to a smaller volume 1. This work is the same (with opposite signs). The work to compress the gas during the illumination can only come from the
illumination. I provided this calculation in my previous report. It is in clear discrepancy with the authors' calculation. This issue needs clarification.
